# Antibacterial and Oxidative Stress-Protective Effects of Five Monoterpenes from Softwood

**DOI:** 10.3390/molecules27123891

**Published:** 2022-06-17

**Authors:** Riina Muilu-Mäkelä, Ulla Aapola, Jenni Tienaho, Hannu Uusitalo, Tytti Sarjala

**Affiliations:** 1Production Systems Unit, Natural Resources Institute Finland, FI-00791 Helsinki, Finland; jenni.tienaho@luke.fi (J.T.); tytti.sarjala@gmail.com (T.S.); 2Department of Ophthalmology, Tampere University, FI-33520 Tampere, Finland; ulla.aapola@tuni.fi (U.A.); hannu.uusitalo@tuni.fi (H.U.); 3Tays Eye Centre, Tampere University Hospital, FI-33520 Tampere, Finland

**Keywords:** antioxidants, antibacterial properties, biosensors, HCE cells, monoterpenes, soft wood material

## Abstract

Volatile organic compounds (VOC) affect the quality of indoor air. Terpenes and especially monoterpenes are the main molecules emitted from softwood material (coniferous species), which is widely used in construction. The corneal epithelium is one of the first human membranes to encounter VOCs in the air. Moreover, the industrial use of pleasant-scented monoterpenes in cosmetics, food, and detergents exposes people to monoterpenes in their daily lives. In the present study, the health effective properties of five monoterpenes from softwood were tested; cytotoxicity and oxidative stress-protective effects of α- and β-pinenes, R- and S-limonene, and 3-carene were tested in a human corneal epithelial (HCE) cell model system and with two additional in vitro antioxidant tests: oxygen radical absorbance capacity (ORAC) and hydrogen peroxide (H_2_O_2_) scavenging. Antibacterial efficacies were tested with two bioluminescent bacterial biosensor strains (*Escherichia coli* K12+pcGLS11 and *Staphylococcus aureus* RN4220+pAT19) and with minimum inhibitory concentration (MIC) test against *Escherichia coli*. Only very high concentrations of monoterpenes (0.3–0.5 mg/mL) demonstrated cytotoxicity against HCE cells. Contrary to the original hypothesis, monoterpenes did not exhibit strong antioxidant properties in tested concentrations. However, biosensors and MIC tests indicated clear antibacterial activities for all tested monoterpenes.

## 1. Introduction

Softwood species such as Scots pine (*Pinus sylvestris* L.) and Norway spruce (*Picea abies* L. Karst.) are commercially important tree species, especially for Northern countries, and serve as the main species of wood construction. Wood material contains natural volatile organic compounds (VOCs), which evaporate to the indoor air and, thus, affect the indoor air quality. Limits for the total VOC contents of construction materials are strictly regulated by health legislation and EU standards [1]. The forthcoming indoor material CE-marking requirements will replace the multitude analyzing methods of total VOC values of materials [2] and might become problematic to wood marketing in the near future. Therefore, better understanding of the health effects of different wood VOCs is needed. On the other hand, for occupational health, the working environment is one of the major factors of working satisfaction and performance [3]. It is also vital for the quality of life of the workers. This is especially important in the modern office work where most of the work is done with computers and displays that are known causes of ocular irritation and ocular surface problems. Due to the ageing population, the environmental factors should also be taken account when designing the spaces where elderly people are spending most of their time. Ageing is also an important factor correlating with the ocular surface disease symptoms and signs [4].

Monoterpenes comprise 80% of the softwood VOCs, and five compounds, α- and β-pinene, 3-carene, and R- and S-Limonene, dominate the compound group [5,6]. The terpenes are a large and diverse class of organic compounds. The structure of terpenes is based on the linkage of isoprene units (C_5_H_8_), such as dimethylallyl pyrophosphate and isopentenyl pyrophosphate [7]. Depending on the number of linked isoprene units, the resulting terpenes are classified into hemi-, mono-, sesqui-, di-, sester-, tri-, sesquar-, tetra-, and polyterpenes. Monoterpenes contain two isoprene units and ten carbon atoms and can form bicyclic (α- and β-pinenes, 3-carene), monocyclic (limonene) or acyclic structures (myrcene).

Limonene, α- and β-pinene, and 3-carene are colorless liquids at room temperature and registered as ‘generally recognized as safe substance’ in the Code of Federal Regulation for synthetic flavorings (U.S. FDA, 2012) [8] (Figure 1). Limonene exists as two optical isomers, named R or d- and S or l-limonene. R-(+)-limonene (d-limonene) is a commonly used flavor additive in food and fragrances for its pleasant lemon-like aroma. (R)-(+)-limonene has a fragrance like fresh citrus, whereas the fragrance of (S)-(−)-limonene is harsh, turpentine-like with a lemon note [9]. Both R and S enantiomers exist in coniferous species as a racemic mixture [10]. Both structural isomers α- and β-pinene have two enantiomers (+) and (−). This difference yields four active isomers, but (+)-β-pinene rarely exist in nature [10,11]. Turpentine is composed primarily of α- and β-pinene and with lesser amounts of 3-carene and other monoterpenes. 3-Carene is a chiral compound, which makes it an interesting molecule for chemical modifications [12].

Monoterpenes show a diverse range of biological activities including antimicrobial, antiparasitic, anti-inflammatory, antioxidant, and antitumoral activities [7]. Oxidative stress is a complex process in biological systems that is characterized by an imbalance between the production of free radicals like reactive oxygen species (ROS) and the metabolic ability to eliminate them [13]. ROS are highly reactive molecules due to unpaired electrons in their structure, and they react with biological macromolecules (carbohydrates, nucleic acids, lipids, and proteins) and alter their functions. In general, detox enzymes (i.e., superoxide dismutase, catalase) are triggered to prevent oxidative reactions and several nonenzymatic antioxidants play a key role in the maintenance of homeostasis. The chemical structure of monoterpenes provides double bonds and reduced functional groups that are susceptible to oxidation and have indicated antioxidant features [14].

There is a correlation between chronic inflammation and oxidative stress. Inflammation is primarily a protective response (e.g., against microorganisms, toxins, or allergens) of the organism to remove the injurious stimuli and to initiate the healing process. However, chronic and uncontrolled inflammation becomes detrimental to tissues [15]. The oxidative damage to cellular components is responsible for several chronic diseases, such as diabetes, asthma, and allergies. Inflammation is also a major contributor to many eye disorders, including uveitis, diabetic retinopathy, and age-related macular degeneration [16].

Cytokines are small, cell-released proteins that have a particular effect on cell–cell interactions and communications. Interleukin 6 (IL-6) is an inflammatory cytokine that is produced rapidly and transiently in response to infections and tissue damage. Inflammatory cytokines promote host defense by stimulating acute phase responses and immune responses. The unregulated continuous synthesis of IL-6 has a pathological effect on chronic inflammation and autoimmunity [17]. Monoterpenes have been shown to have a promising profile as substances that reduce the inflammatory process and modulate key chemical mediators in the inflammatory cascade, such as cytokines [18]. VOCs derived from conifer species have been shown to produce anti-inflammatory effects by regulating cytokine production in mouse lung cells [19]. Moreover, limonene has been shown to protect human lens epithelial cells in cell culture under oxidative stress conditions [20].

Terpenes have antibacterial properties, which are based on the ability of molecules to act on microbial cell walls [21,22]. The antibacterial properties of the wood material are based, at least in part, on the compounds in the wood, such as terpenes [23]. The ability of materials to influence the growth of microbes also has an effect on indoor air quality, e.g., reducing toxins excreted by microbes. Although the high concentrations of VOCs from wood material can irritate the epithelium of the eyes or nose [24], the terpene concentrations in the indoor air of wooden buildings tend to remain low and the odor of wood is often considered pleasant [25].

Generally, VOCs emitted from construction materials are measured as the total VOC content and the health effect of individual compounds are not well-known. This study examines the importance of typical wood compounds for eye health and more generally for well-being. The outermost layer of the eyes is formed by human corneal epithelial cells (HCE) that are covered by a thin tear fluid layer [26]. HCE cells are the first cells that face the VOCs of indoor air. Several of the most common monoterpenes, with α-pinene and 3-carene as the major ones, emitted from pine and spruce construction products [27] are also inhaled during forest walking [28]. Thus, in the present study, oxidative stress-protecting properties and potential cytotoxicity levels of most common monoterpenes from softwood were investigated in the HCE liquid cell cultures [29]. Different concentrations of monoterpenes were added to the HCE cell cultures and the viability of the eye cells as well as IL-6 production was monitored. In addition, the aim of the study was to evaluate the antioxidant and antibacterial features of monoterpenes using two bioluminescent bacterial biosensor strains, MIC test and antioxidant tests (oxygen radical absorbance capacity (ORAC_FL_) and H_2_O_2_ scavenging assay). Of course, the eye cell model does not correspond to a real situation where the HCE cells are part of the eye. However, the model can be used to find limit values for effective concentrations. Finally, the direct effects of monoterpenes α- and β-pinenes, R- and S-limonene, and 3-carene on eye well-being are discussed, but also the effects of terpenes on microbes, whose growth inhibition, among other things, has a positive effect on indoor air.

## 2. Results

### 2.1. Cytotoxic and Oxidative-Stress Protective Effects of Monoterpenes in HCE-Cell Cultures

Cytotoxicity of all the five monoterpenes were tested with concentration range 0–5 mg/mL by using WST-1 test. Monoterpene concentration lower than 0.3 mg/mL did not affect the viability of HCE cells (Figure 2). Decline of HCE cell viability was observed at concentrations 0.3–0.5 mg/mL of α-pinene, 3-carene, R-limonene, and S-limonene. β-Pinene was not toxic to HCE cells. The half maximal inhibitory concentrations (IC50) of investigated monoterpenes are listed in Table 1.

According to the cytotoxicity results, the concentrations 0.1 and 0.5 mg/mL of each terpene were chosen for the following analyses. At the concentration 0.1 mg/mL, none of the tested monoterpenes affected the viability of the HCE cells. However, a clear decline of cell viability was seen at the concentration 0.5 mg/mL of S- and R-limonene, α-pinene, and 3-carene, and this concentration was chosen for the oxidative stress cell model to indicate the maximal protective effect. In the cell culture model for oxidative stress, HCE cells were exposed to 800 µM H_2_O_2_, with and without monoterpenes. The H_2_O_2_ addition decreased the viability of the HCE cells to less than 10%, as was expected (Figure 3). Monoterpenes did not protect the HCE cells against oxidative stress, except α-pinene, which indicated some protecting features at the concentration 0.1 mg/mL (Figure 3).

The inflammation marker protein IL-6 production was measured at two different exposure times from the cell culture media (Figure 4). At the concentration 0.5 mg/mL, α-pinene and 3-carene induced a strong IL-6 production, which indicated a similar inflammation response as caused by lipopolysaccharide (LPS) used as a positive control. The other terpenes had no effect on IL-6 production of the HCE cells. The strong H_2_O_2_ treatment with or without monoterpenes did not affect the IL-6 production either, probably because the dose was lethal (Appendix A) [30].

### 2.2. Antioxidant Properties of Monoterpenes Measured by ORAC and H_2_O_2_ Scavenging Assays

Antioxidant properties of terpenes were tested with H_2_O_2_ scavenging (also a modified ferrous oxidation–xylenol orange (FOX) reagent method) and oxygen radical absorbance capacity (ORAC) tests. S-Limonene and 3-carene showed the highest ORAC activities of the tested terpenes (Figure 5). Instead, the tested monoterpenes had very low hydrogen peroxide scavenging activity (H_2_O_2_ inhibition: 2.4–5.4%; data not shown).

### 2.3. Antibacterial Efficacy of Monoterpenes Measured by MIC-Test and Biosensors (E. coli K12+pcGLS11 and S. aureus RN4220+pAT19)

The minimum inhibitory capacity (MIC) test indicated that for α-pinene, β-pinene, 3-carene, and R-limonene, the bacterial growth inhibiting contents against *E. coli* were >1.25 mg/mL, >1.25 mg/mL, >0.078 mg/mL, >0.63 mg/mL, respectively (see Appendix A). The terpene 3-carene has an unusual shape in its absorbance curve. This phenomenon occurs because it absorbs the used wavelength of 600 nm, especially in the higher concentrations (data not shown).

According to the results of MIC test, three different concentrations (0.82, 1.6, and 3.2 mg/mL) of α- and β-pinene, 3-carene, and S-limonene were tested with two biosensor strains *E. coli* K12+pcGLS11 and *S. aureus* RN4220+pAT19, both first reported by Vesterlund et al. [31]. All the tested terpenes indicated dose-dependent responses, and 3-carene and S-limonene were the most effective inhibitors. With 3-carene and S-limonene, even the lowest concentration (0.82 mg/mL) effectively inhibited the luminescent light production and cellular viability of both biosensor strains (Figure 6 and Figure 7). The highest content of the solvent dimethyl sulfoxide (DMSO, 6.3 % found in 3.2 mg/mL of monoterpene concentrations) was almost as effective as the highest concentration of β-pinene (3.2 mg/mL) with both biosensor strains. After 50 min incubation, the effects of different terpene concentrations on light production of the bacterial strains were statistically analyzed and the obtained results are presented in Appendix A.

## 3. Discussion

A high indoor air VOC content evaporated from construction materials can cause eye irritation and other eye symptoms [25,32]. On the other hand, volatile monoterpenes have been shown to have therapeutic potential for many inflammatory diseases [7]. In the present study, the effects of the five monoterpenes on cell viability and their potential ability to protect cells from oxidative stress were tested in liquid HCE cell cultures. Contrary to expectations, the results did not indicate strong protective effect of monoterpenes (α- and β-pinene, 3-carene, and S- and R-limonene) against oxidative stress in HCE cultures. Additionally, the low H_2_O_2_ inhibition (2.4–5.5% inhibition) observed in the H_2_O_2_ scavenging assay was in line with the results from HCE experiments. Furthermore, low H_2_O_2_ scavenging activity was comparable to, for example, needle and shoot extracts of spruce [33] which may suggest that monoterpenes are not strong protective molecules against H_2_O_2_-induced oxidative stresses. However, D-limonene has previously been shown to reduce H_2_O_2_-induced cell death in human lens epithelial cells (HLEC) and was safe for cells until the concentration reached 0.27 mg/mL [20]. In the present study, the noxious concentration of R-limonene was approximately 0.4 mg/mL, but no cell protective effect was observed. Instead, a small protective effect against oxidative stress was observed with 0.1 mg/mL α-pinene. Similarly, in a previous study with human astrocytoma cell cultures, α-pinene treatment was shown to inhibit ROS formation and lipid peroxidation and to protect cells from H_2_O_2_-stimulated oxidative damage [34]. In addition, UVA radiation has been suggested to increase peroxidation and ROS formation, which, however, is inhibited by α-pinene pretreatment [35].

The ORAC values indicated higher antioxidative activity of S-limonene and 3-carene than either of the pinenes or R-limonene. However, in general, the ORAC values of monoterpenes were lower than, for example, those of pure stilbenes, a group of highly antioxidative compounds in trees [36]. In comparison, by using the same ORAC assay protocol the antioxidative power of resveratrol, a known antioxidative stilbene derivative, showed about 10 times higher ORAC value than 3-carene or S-limonene in the present study.

IL-6 is often used as a marker protein to indicate launched inflammation procedures in tissues [17]. In HCE cultures, both α-pinene and 3-carene were cytotoxic and induced IL-6 production at the concentration of 0.5 mg/mL. In a previous study, the highest 3-carene emission was 0.00003 mg/mL from the fresh Scots pine blocks [6]. Very high content of terpenes can be harmful for eyes, but according to the present experiment, the cytotoxicity level is thousands of times higher than that content evaporated from the indoor wood material [5,6,25]. Out of the five tested monoterpenes, β-pinene did not indicate any cytotoxic properties in HCE cell cultures. In the future, the obtained results on the effects of wood emissions on eye function could be confirmed, for example, by proteomics studies [4].

A wide range of pharmacological activities of α- and β-pinene have been summarized [37]. For example, α-pinene is considered as an antimicrobial compound, which is shown to be effective against food-borne pathogens, such as *Bacillus cereus*, *Escherichia coli*, and *Campylobacter jejuni* [38]. In the present study, concentrations of monoterpenes showing antibacterial properties were slightly higher than cytotoxicity concentrations in the HCE cell model. Biosensor measurements indicated antibacterial features of α-and β-pinene that inhibited the light production of *E. coli* (K12+pcGLS11) and *S. aureus* (RN4220+pAT19) biosensor strains. However, S-limonene and 3-carene indicated even stronger activity in the biosensor experiments than the pinenes.

Potentially the used enantiomers of pinenes affected their antibacterial efficiency. It has been reported that only positive enantiomers of α- and β-pinene are effective against microbes [39]. In the present study (−)-β-pinene was used because it is more abundant in nature and in essential oil of Pinus and Picea than (+)-β-pinene [10]. Both α-pinene enantiomers (+) and (−) exist in wood, and, therefore, in this study, a mixture of both enantiomers was tested.

In previous studies, it has been concluded that the antibacterial efficacy of d-limonene (=R-limonene) against *Listeria monocytogenes* [21], as well as 3-carene against Gram-positive *B. thermosphacta* and Gram-negative *P. fluorescens* [22], is caused by the ability of monoterpenes to cause damage to cell membranes [21]. Furthermore, it was concluded that both limonene and 3-carene affect respiration and energy metabolism by inhibiting the function of the respiratory chain complex, which evidentially led to the cell death of bacteria [21,22]. Limonene has been utilized in food preservation, but its efficacy is decreased because many microbial strains are able to develop resistance against it.

In the present study, the antimicrobial effects measured by MIC and bacterial biosensors indicate that the monoterpenes play a role in the antibacterial effects of wooden surfaces, which is also supported by previous literature [23]. More investigations are needed to evaluate the role of terpenes and other wood extractables against harmful indoor environment microbes, such as molds and bacteria, and to find a balance between antimicrobial functionality and the total VOC contents in the indoor air quality point of view. In addition to the eye health, any potential antimicrobial effects of the indoor volatile monoterpenes on respiratory tracts through inhalation would be important to acknowledge when evaluating the overall health effects of VOCs of materials.

## 4. Materials and Methods

### 4.1. Cell Model Tests: HCE Cell Model and IL-6

Protective and detrimental effects of terpenes on human corneal epithelial cell (HCE) cultures were studied with the human corneal epithelial cells (HCE) model [29]. HCE liquid cell cultures were exposed to five terpenes, (±)-α-pinene 98% (#147524), (−)-β-pinene 99% (#112089), 3-carene analytical standard (94415), (R)-(+)-limonene analytical standard (#62118-1ML), and (S)-(−)-limonene analytical standard (#62128-1ML) (Sigma). Dimethyl sulfoxide (DMSO) was used to dilute hydrophobic terpenes. Due to the cytotoxic feature of DMSO, the effect of DMSO on HCE cells was separately examined by using the concentrations that were used to dilute the terpenes. Dilution series of all five terpenes were made with contents of 0.001, 0.01, 0.1, 1, 10, 100, 500, 1000, and 5000 μg/mL (Appendix A). According to the results, in the dilutions with <1000 µg/mL of terpenes, the DMSO content had no effects on cellular viability, whereas DMSO in >1000 µg/mL terpene dilutions was cytotoxic by itself.

The HCE cells were grown in Dulbecco’s MEM/Nut MIX F-12 medium (1:1) (Gibco), supplemented with 15% (*v*/*v*) FBS (Gibco), 1% (*v*/*v*) antibiotic solution (penicillin 10,000 U/mL, streptomycin 10,000 μg/mL, and amphotericin B 25 μg/mL; Gibco), 2 mmol/L L-glutamine (Gibco), and 5 μg/mL insuline (Sigma, Saint Louis, MO, USA) and 10 ng/mL EGF (Sigma, Saint Louis, MO, USA). All the cell model tests were made on 96-well plates using 30,000 cells/well.

First the cytotoxic properties of terpenes were analyzed by pinpointing the harmful concentration of the individual terpenes. Different concentrations (0–5000 μg/mL) of terpenes were added to the cell cultures and the viability of the corneal cells were observed by a WST-1 assay (Roche). WST-1 assay is based on functions of mitochondrial dehydrogenase enzymes as an indication of cellular growth and viability. The WST-1 cell proliferation reagent indicates the ability of cells to degrade tetrazolium salt to formazan which only happens in the metabolically active cells. After 24 h at 37 °C cells were exposed to 0–5000 μg/mL terpenes and incubated for another 24 h. After 24 h incubation, cytotoxicity was evaluated with the WST-1 assay (Roche) and the absorbances were measured by Victor Multilable Counter according to manufacturer’s instructions. Measurements were repeated three times with six parallel samples and the exposing media did not contain serum (FBS). The half-maximal inhibitory concentration IC50 values were calculated using the R package ‘dr4pl’, which implements the 4 Parameter Logistic model.

Hydrogen peroxide (H_2_O_2_) was used to induce oxidative injury to HCE cells. First, the most effective H_2_O_2_ content (200–800 μM) was screened and 800 µM killed the cells effectively. HCE cells were exposed to 800 μM H_2_O_2_ with and without 0.1 or 0.5 mg/mL monoterpenes. For the H_2_O_2_ measurements, three parallel samples were used. The results were calculated by comparing treated and non-treated samples together and by setting the viability value of non-treated samples on the level 100%.

IL-6 was measured from the cell culture media by using BD OptEIA™ Human IL-6 ELISA kit (BD Biosciences) according to manufacturer’s instructions. Lipopolysaccharide (LPS) is an endotoxin and induces IL-6 expression and was used as a positive control in the tests.

### 4.2. In Vitro Antioxidant Assays: ORAC and H_2_O_2_ Scaveging

Antioxidant properties of the monoterpenes were assessed with a hydrogen atom transfer based ORAC (Oxygen radical absorbance capacity) and H_2_O_2_ scavenging assays. The compounds were dissolved in 99.5% ethanol with concentrations from 5 mg/mL to 37.5 mg/mL to fit the sample concentrations on an appropriate level.

The Oxygen Radical Absorbance Capacity (ORAC_FL_) assay is a hydrogen atom transfer-based method, which measures the oxidative dissociation of fluorescein at the presence of peroxyl radicals (R-O-O) causing reduction in the fluorescence signal. The antioxidant’s protective ability is based on the inhibition of the breakdown of fluorescein in the reaction mixture caused by the peroxyl radicals. The assay was modified from the method described by Huang et al. [40] and Prior et al. [41] and carried out in a 96-well format with two technical replicates of each sample on the plate. Each reaction mixture contained 25 μL of the sample in 0.075 M phosphate buffer pH 7.5 (Merck), 150 μL of 8.16 × 10^−5^ mM fluorescein (Sigma-Aldrich Chemie GmbH, Steinheim, Germany), and 25 μL of 2,2′-Azobis (2-methylpropionamidine) dihydrochloride (Sigma-Aldrich Chemie GmbH, Steinheim, Germany). For each sample, a protocol with a series of five dilutions (1:1–1:320) was used and additional dilutions if needed to adjust the sample concentration to the standard curve. An aliquot of 0.153 mM Trolox ((±)-6-Hydroxy-2,5,7,8-tetramethylchromane-2-carboxylic acid, vitamin E analog) (Sigma-Aldrich Chemie GmbH, Steinheim, Germany) was used as the standard compound and the results are expressed as Trolox equivalents (μmol TE/g). Vitamin C (L(+)-ascorbic acid; Merck KGaA, Darmstadt, Germany) was used as a reference compound. Multiple pairwise-comparisons between means of ORAC-values of groups were performed by using one-way ANOVA and Tukey’s test in R environment version 4.0.4.

The hydrogen peroxide (H_2_O_2_) scavenging activity, based on transition metal chelation, was determined by using a method modified from Hazra et al. [42] and Jiang et al. [43] with a microplate reader in a 96-well format with four technical replicates on each plate. An aliquot of 2 mM H_2_O_2_ (Merck KGaA, Darmstadt, Germany) was added to the reaction mixture with the sample, 2.56 mM ammonium iron (II) sulphate 6H_2_O (BDH Prolabo) and 111 μM xylenol orange disodium salt (Sigma-Aldrich Chemie GmbH, Steinheim, Germany). After 30 min incubation, the absorbance of ferric-xylenol orange complex at 560 nm was measured. The assay measures the ability of the sample to scavenge H_2_O_2_ and prevent the oxidation of Fe (II) to Fe (III) which is indicated by the formation of ferric-xylenol orange complex. The H_2_O_2_ scavenging ability is expressed as inhibition percentage (%) of Fe (II) oxidation to Fe (III). Sodium pyruvate (Sigma-Aldrich Chemie GmbH, Steinheim, Germany) was used as a reference compound.

### 4.3. Antibacterial Efficacy

Minimum inhibitory concentration (MIC) is the lowest concentration of a chemical which prevents the growth of bacteria. MIC-test indicates the lowest concentration of a monoterpene necessary to inhibit the visible growth of bacteria. The used Top10 chemically competent *E. coli* strain (Invitrogen) was stored at −80 °C before overnight cultivation until A_600_ values reach 0.07–0.1, which is equivalent of 10^4^ to 10^5^ CFU/mL. The MIC test was performed with a standard microtiter broth dilution method, where first, an aliquot of 50 µL of Mueller–Hinton broth (Merck KGaA, Darmstadt, Germany) was pipetted into every well of a clear polystyrene 96-well plate. The Mueller–Hinton broth was prepared according to the manufacturer instructions (21 g of powder into 1 L of double-distilled water and 15 min autoclaving at 121 °C; pH 7.4). Ten increasing concentrations of the monoterpenes (±)-α- and (−)-β-pinene, 3-carene and R-(+)-limonene dissolved in DMSO (10–5000 µg/mL) were then prepared and pipetted in aliquots of 50 µL into the 96-well plate in 4 technical replicates (shown as A–D in the Appendix A). Finally, 5 µL of the bacterial cultivation was added into all but the blank or positive control wells of the microplate. Thus, positive control or sterility wells included no bacterial cultivation and negative controls only bacterial cultivation with the Mueller–Hinton broth. The measurement plate was then incubated in 37 °C with a 150-rpm shaking for approximately 16 h and then measured for absorbance at 600 nm with a Varioskan Flash Multilabel device (Thermo Fischer Scientific, Thermo Electron Co., Waltham, MA, USA). The results are expressed in concentrations (µg/mL), where no bacterial growth is detected.

A microplate method with bioluminescent indicator strains *Staphylococcus aureus* RN4220+pAT19 and *Escherichia coli* K12+pcGLS11 [31] was used to study the antibacterial activity of the α- and β-pinene, 3-carene and R- and S-limonene. These strains have been constructed to produce a constant luminescent light signal and antibacterial effects can be observed as a loss of emitted light signal intensity. The storage, cultivation, and test protocol has been previously described [36,44]. In brief, the strains were stored at −80 °C and awoken by approximately 16 h cultivation at 30 °C (for *E. coli*) and 37 °C (for *S. aureus*) in lysogeny agar plates (tryptone 10 g/L; yeast extract 5 g/L; NaCl 10 g/L; and agar 15 g/L). The LA plates were supplemented with 10% (*v*/*v*) sterile filtered phosphate buffer (PB) (1 M, pH 7.0) and 100 µg/mL of ampicillin for *E. coli* and with 5 µg/mL erythromycin for *S. aureus*. Biosensor stocks were prepared by inoculating a single colony of bacteria in lysogeny broth with same supplements as plates. Stocks were cultivated for approximately 16 h at 300 rpm shaking at 30 °C (*E. coli*) and 37 °C (*S. aureus*). The monoterpenes (±)-α- and (−)-β-pinene, 3-carene, and R-(+)-limonene were first dissolved in DMSO and diluted with water to obtain the terpene concentrations 3.2, 1.6, and 0.82 mg/mL per microplate well with the highest DMSO content of 6.3%. The highest concentrations of terpenes formed a minor precipitation with water, but it did not seem to affect the results. The monoterpenes and positive (DMSO, ethanol) and negative (double-distilled water) controls were pipetted in triplicates into opaque white polystyrene microplates with same volume of bacterial inoculation. The produced luminescent light signal was then measured using a Varioskan Flash Multilabel device (Thermo Fischer Scientific, Thermo Electron Co., Waltham, MA, USA) once every 5 min for 95 min at room temperature, and the plate was briefly shaken before every measurement. The results are expressed as relative light units (RLUs) drawn as a function of the incubation time. Error bars represent the standard deviations between the sample triplicates. The effects of different terpene concentrations on light production of both bacterial strains were statistically analyzed by using one-way ANOVA and Tukey’s multiple comparisons of mean in R environment version 4.0.4. The incubation time 50 min was chosen to be the most descriptive time point to compare differences.

## 5. Conclusions

In everyday life, people are exposed to monoterpenes, which evaporate from wood material and are also used in detergents and food additives. Many health-enhancing effects of monoterpenes have been reported, while some studies emphasize their allergenic and irritant properties. Corneal epithelium cells are the first cells in the human eye to encounter compounds from the indoor air. Here, we indicate that the harmful level of monoterpenes, α- and β-pinene, R- and S-limonene, and 3-carene for ocular epithelial cells is thousands of times higher than the levels known to be released from wooden construction materials. Therefore, the indoor air terpene contents evaporated from wooden structures are safe for eye epithelial cells. Interestingly, slight indication of α-pinene oxidative stress-protective effects was observed, which should be further studied in the future. In addition, all five monoterpenes were effective against the growth of both Gram-positive and Gram-negative bacteria as detected by the MIC test and two microbial biosensor strains. The results of this study indicate that the roles of α- and β-pinene, R- and S-limonene, and 3-carene might be more important in inhibiting microbial growth than in protecting cells and tissues from the oxidative stress. Concentrations of wood-volatile monoterpenes are not particularly harmful to ocular epithelial cells, but, also, no clear health effects were observed in this study. The strong antimicrobial properties of monoterpenes appear to be the most significant factor influencing indoor air quality in wooden environments.

## Figures and Tables

**Figure 1 molecules-27-03891-f001:**
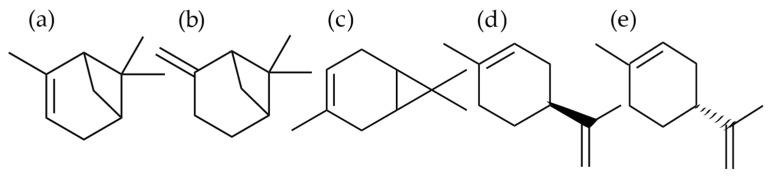
Molecular formula of (**a**) α-pinene, (**b**) β-pinene, (**c**) 3-carene, (**d**) S-limonene, and (**e**) R-limonene. Monoterpenes α- and β-pinene are structural isomers, and R- and S-limonene are two enantiomers found in nature, e.g., in pine (coniferous trees).

**Figure 2 molecules-27-03891-f002:**
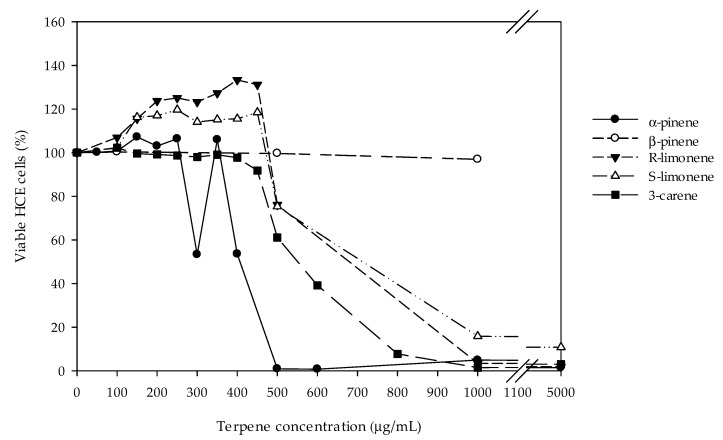
Cytotoxicity results for α- and β-pinene, R- and S-limonene, and 3-carene in the HCE cell model measured with a WST-1 test. The values represent average cell viabilities from one to three repeated measurements with monoterpene concentrations of 0–5000 µg/mL. Results obtained with control cells were set as 100%.

**Figure 3 molecules-27-03891-f003:**
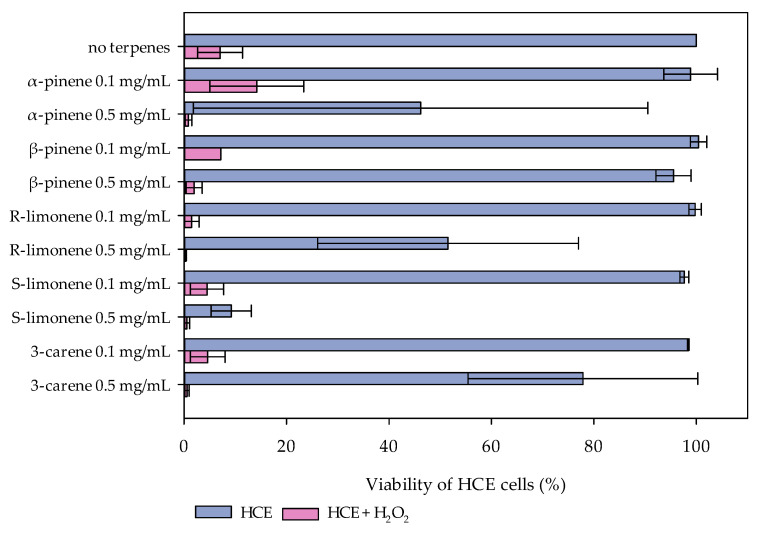
The oxidative stress-protective effect of α- and β-pinene, R- and S-limonene, and 3-carene with two different concentrations (0.1 and 0.5 mg/mL) in the HCE cell model. The values represent the average and range of the HCE cell viability (%) (sample size = 2) without (light blue bars) and with (light red bars) hydrogen peroxide (800 µM H_2_O_2_) treatment. Monoterpenes did not show a clear oxidative stress-protection for the HCE cell cultures.

**Figure 4 molecules-27-03891-f004:**
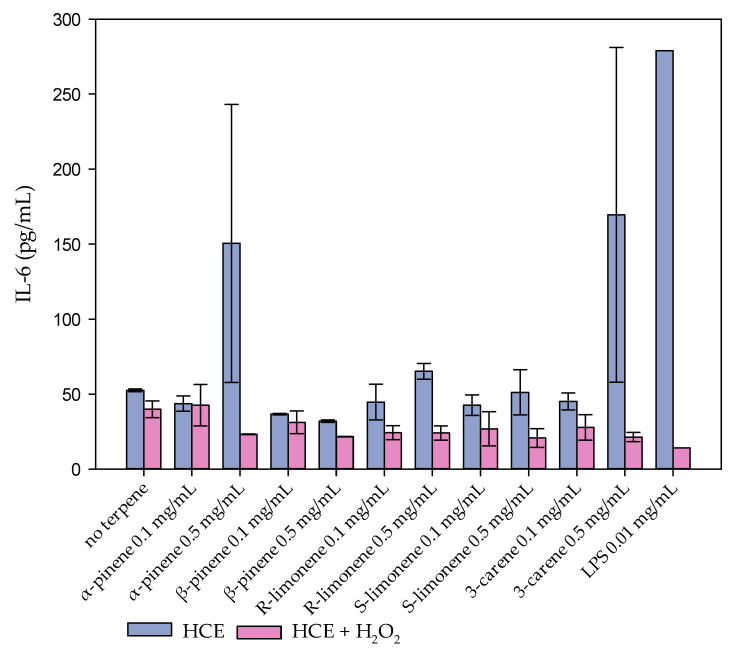
Inflammation marker protein (IL-6) was induced under 0.5 mg/mL α-pinene and 3-carene treatments. The values represent the average and range (sample size = 2) of IL-6 marker content (pg/mL) in the HCE cell cultures that were treated with 0.1 or 0.5 mg/mL α- or β-pinene, R- or S-limonene, or 3-carene without (light blue bars) and with H_2_O_2_ treatment (light red bars). Lipopolysaccharide (LPS) was used as a positive control.

**Figure 5 molecules-27-03891-f005:**
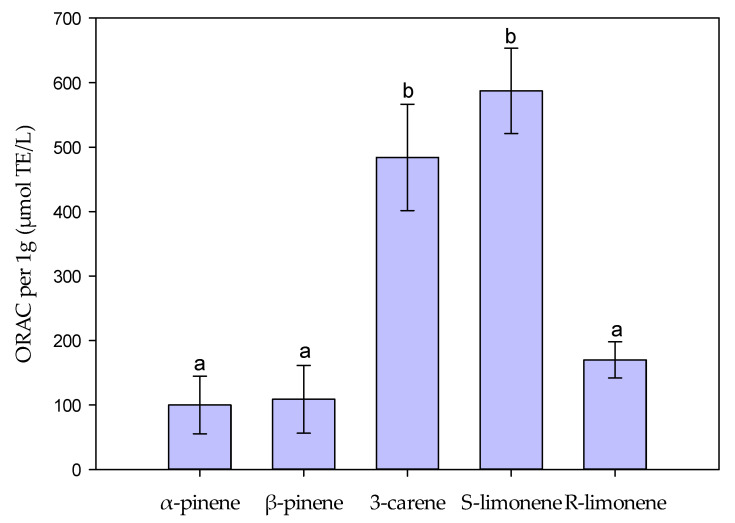
Oxygen radical absorbance capacity (ORAC) results of 1 g of α- and β-pinene, 3-carene, and S- and R-limonene. 3-Carene and S-limonene indicated the highest antioxidant activities of the tested monoterpenes. The values represent the µmol Trolox Equivalent (TE) averages ± standard deviations (sample size = 4) from 1 g of monoterpenes. The letters above the bars indicate significant differences between the means of the groups (*p* value < 0.05) evaluated by one-way analysis of variance (ANOVA) completed with Tukey’s test.

**Figure 6 molecules-27-03891-f006:**
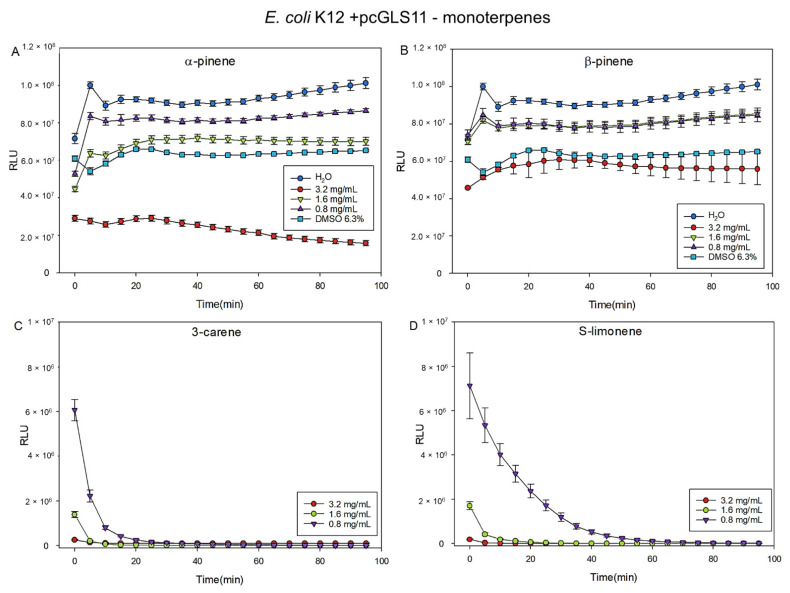
Effect of the monoterpenes (**A**) α-pinene, (**B**) β-pinene, (**C**) 3-carene, and (**D**) S-limonene for the viability of continuously luminescent light producing biosensor strain *E. coli* K12+pcGLS11. The highest content of DMSO 6.3% found in the highest concentration (3.2 mg/mL) of monoterpenes is drawn as a control in A and B. The figure shows the relative light unit (RLU) averages of the triplicates in the microplate. Lower the RLU aka light production values, the more efficient the terpene in question is in inhibiting the bacterial metabolism, which indicates antibacterial activity. The error bars represent the standard deviations of the sample triplicates.

**Figure 7 molecules-27-03891-f007:**
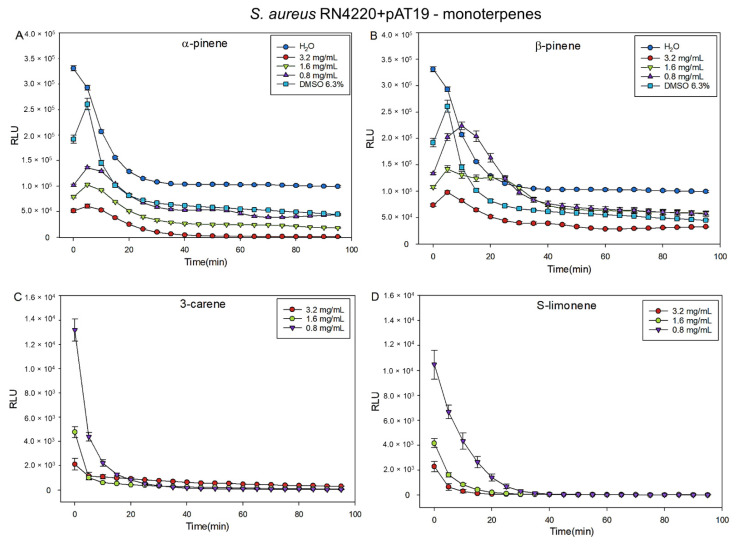
Effect of the monoterpenes (**A**) α-pinene, (**B**) β-pinene, (**C**) 3-carene, and (**D**) S-limonene for the viability of continuously luminescent light producing biosensor strain *S. aureus* RN4220+pAT19. The highest content of DMSO 6.3% found in the highest concentration (3.2 mg/mL) of monoterpenes is drawn as a control in A and B. The figure shows the relative light unit (RLU) averages of the triplicates in the microplate. Lower the RLU aka light production values, the more efficient the terpene in question is in inhibiting the bacterial metabolism indicating antibacterial activity. The error bars represent the standard deviations of the sample triplicates.

**Table 1 molecules-27-03891-t001:** The half-maximal inhibitory concentration (IC50) of monoterpenes.

Terpene	IC50
α-pinene	401
β-pinene	nd ^1^
R-limonene	502
S-limonene	502
3-carene	552

^1^ nd, not detectable in tested concentration range.

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
