# Peer review of "Antibacterial and Oxidative Stress-Protective Effects of Five Monoterpenes from Softwood"

_molecules, 2022, doi:10.3390/molecules27123891_

Round 1

Reviewer 1 Report

The manuscript "Antibacterial and oxidative stress-protective effects of five  monoterpenes from softwood” by Muilu-Mäkelä, R et al. describe the study on the health effective properties of five monoterpenes from softwood. The cytotoxicity, oxidative stress-protective effects and antibacterial efficacies of α- and β-pinenes, R- and S-limonene and 3-carene were tested. Overall the work has been carried out with high scientific standards and the presentation of the results is correct. The study was documented appropriately and have been performed according to published procedures. The research has shown that the examined monoterpenes did not exhibit strong antioxidant properties but biosensors and MIC tests indicated antibacterial activities.

I have no major suggestions to improve the quality of the manuscript, therefore I suggest its publication to Molecules in the present form.

Author Response

We thank you for the encouraging comments and for a suggestion to accept our paper to be published in Molecules. We have made some revisions on our manuscript such as statistical analysis and we have illustrated more the importance of the study. Please find the manuscript attached on the system.

Reviewer 2 Report

The manuscript entitled, ‘Antibacterial and oxidative stress-protective effects of five monoterpenes from softwood’ studies the antioxidative and antibacterial features of monoterpenes with two bioluminescent bacterial biosensor strains and antioxidative activities of five monoterpenes, α- and β-pinenes, R- and S- limonene, and 3-carene. However, the authors should clarify some major concerns before any possible consideration of this manuscript to be published in the journal of ‘molecules’.

  • The authors must describe the novelty of their study in the ‘Introduction’ and ‘Conclusion’ section?
  • What is the practical application of this study in the welfare of mankind? The bottlenecks in this type of studies should be discussed in the ‘Introduction’ section.
  • The authors should illustrate more for a better understanding of the ‘Discussion’ part and to make this manuscript interesting.
  • The authors should prepare suitable figures for the scientific paper. Most of the figures of this manuscript are not well organized. For example: the figures of this manuscript have wrong aspect ratios. Figure legends are not well placed and Figure captions are insufficient.
  • Why there is no error bars in some figures of this manuscript?
  • Data related to statistical significance is necessary to support the findings of this study.
  • The authors can present the pictorial evidence for the antibacterial activity experiments such as plate assay, SEM of TEM analysis.
  • The importance of this manuscript’s findings must be included in the ‘Conclusion’ section.

Author Response

We thank for the valuable comments, which have helped us to improve the manuscript. Herewith you can find the step by step answers to all comments. The manuscript has been revised according to suggestions.

Reviewer 2:

ï‚·  The authors must describe the novelty of their study in the ‘Introduction’ and ‘Conclusion’ section?

Thank you for your valuable comment, we added few sentences to highlight the novelty of the research. Please, see the page 3 and lines 111-114 and 130-133.

lines 111-114: Volatile compounds emitted from building materials are generally treated as total VOCs and the effect of individual compounds on health are not well known. This study examines the importance of typical wood compounds for eye health and generally for well-being.

lines 130-133: Finally, the direct effects of monoterpenes α- and β-pinenes, R- and S-limonene and 3-carene on eye well-being are discussed, but also the effects of terpenes on microbes, whose growth inhibition has, among other things, a positive effect on indoor air quality.

ï‚·  What is the practical application of this study in the welfare of mankind? The bottlenecks in this type of studies should be discussed in the ‘Introduction’ section.

To describe better the practical application of this study we decided to start introduction with the sentences:

Page 1, lines 40-47: For occupational health the working environment is one of the major factors of working satisfaction and performance. It is also vital for the quality of the life of the workers. This is especially important in the modern office work where most of the work is done with computers and displays that are known causes of ocular irritation and ocular surface problems. Due to the ageing population the environmental factors should also take account when designing the spaces where elderly people are spending most of their time. Ageing is an also an important factor which is correlating with the ocular surface disease symptoms and signs.

To the bottleneck question we have answered by adding the sentences:

line 127-130: The eye cell model does not correspond to a real situation where the cells are part of the eye. However, the model can be used to find limit values ​​for effective concentrations. Likewise, antibacterial tests and antioxidant tests are indicative determinations of the efficacy of compounds.

ï‚·  The authors should illustrate more for a better understanding of the ‘Discussion’ part and to make this manuscript interesting.

Thank you, we agree, and the Discussion and introduction were improved considering this comment.

ï‚·  The authors should prepare suitable figures for the scientific paper. Most of the figures of this manuscript are not well organized. For example: the figures of this manuscript have wrong aspect ratios. Figure legends are not well placed and Figure captions are insufficient.

The aspect ratios of the figures were rechecked and the figures 6 and 7 were revised. The size of the figures was equalized, and the Figure captions corrected as suggested.

ï‚·  Why there is no error bars in some figures of this manuscript?

There are no error bars in Figure 2 because we wanted to make the figure as clear as possible. Near the time when viability collapses, there is a big deviation. Before and after collapsing the deviation is very small. We wanted to draw all the monoterpenes on the same pattern and with the error bars this would look messy. So, we prefer not to use errorbars in figure 2. All other figures have errorbars, that illustrate range or sd of the observations.

ï‚·  Data related to statistical significance is necessary to support the findings of this study.

Thank you, we agree, and statistics were included as follows:

In figures 3 and 4 the number of observations is only two, as was written in the figure captions. Therefore, the statistical analyses do not make sense in those cases.

In figure 5 there is a significant difference in ORAC values between 3-carene and S-limonene compared to pinenes and limonene. The results of the one-way variance analysis (ANOVA) with Tukey´s comparisons are now indicated with letters above the bars in the figure 5 and the results are listed on the supplementary Table S1.

Similarly, one-way variance analysis was done for the figures 6 and 7. The incubation time 50min was chosen to describe the differences between the treatments. The results of the statistics from the biosensor measurements are listed in Supplementary Tables S2-S9.

ï‚·  The authors can present the pictorial evidence for the antibacterial activity experiments such as plate assay, SEM of TEM analysis.

Antibacterial effects are presented as numerical values ​​from the MIC and two different biosensor tests. We think that the numeric results are adequate and somewhat more accurate than the images of the bacterial discs would be. We hope that the decision is also right for the reviewer.

ï‚·  The importance of this manuscript’s findings must be included in the ‘Conclusion’ section.

The sentence was added into conclusion to enhance the importance of the study. Please see the Page 13 and Lines 445-448: Concentrations of wood-volatile monoterpenes are not particularly harmful to ocular epithelial cells, but also no clear health effects were observed in this study. The strong antimicrobial properties of terpenes appear to be the most significant factor influencing indoor air quality in wooden environments.

Reviewer 3 Report

The manuscript from Muilu-Makela et al. is an in vitro study investigating the toxicity, antioxidant and antibacterial effect of five volatile monoterpenes released from softwood.

These bioactive properties are investigated by chemical tests as well as in vitro cellular tests using the human corneal epithelial cell model, since these cells are among the first coming to contact with these volatile compounds released in the indoor air of wooden buildings.

The data presented are interesting, but some criticisms should be addressed before publication of the manuscript.

1)     The English needs to be revised.

2)     In the description of the results of Fig. 2 (page 4, lines 130-139) it would be useful to indicate the IC50 of each compound tested on the HCE model, since it can be easily calculated from the cytotoxicity test already performed. This may help to make comparisons with other substances of the same family or belonging to the same group of volatile organic compounds released from softwood.

3)     Why did the authors used such a high concentration of the H2O2 oxidative stimulus? I would have tested the ability of the compounds to counteract the oxidative stimulus in milder conditions, for example with a H2O2 concentration causing a 50% cell toxicity instead of 90%. Can they explain this choice?

4)     Statistical analyses are missing in all graphs. Authors should add an adequate statistical analysis to properly highlight differences among the various compounds.

5)     For what concerns section 4, Materials and Methods, please create separate subsections for the various experiments and avoid long descriptions of the principles of the methods used, just the details of the experiments performed to replicate them.

6)     In the Conclusions section (page 12, lines 388-390), the authors state that “we show that the known evaporated amount of monoterpenes ….. released from wooden construction materials is thousand times lower than the harmful level for ocular epithelial cells”, however there is no reference to the literature that allow them to perform such comparison. I would suggest to move these sentence from the conclusion and add it to the discussion to develop it better and explain how the authors are able to say that the monoterpenes tested are not toxic at the concentrations released from softwood based on the tests performed in their manuscript.

Author Response

We thank for the valuable comments, which have helped us to improve the manuscript. Herewith you can find the step by step answers to all comments. The manuscript has been revised according to suggestions.

The English needs to be revised.

The English was revised by an expert in English language.

2)     In the description of the results of Fig. 2 (page 4, lines 130-139) it would be useful to indicate the IC50 of each compound tested on the HCE model, since it can be easily calculated from the cytotoxicity test already performed. This may help to make comparisons with other substances of the same family or belonging to the same group of volatile organic compounds released from softwood.

Thank you of your valuable comment, we have added a table with the IC50 values below the figure 2 to make it easier to compare the effects of different compounds.

3)     Why did the authors used such a high concentration of the H2O2 oxidative stimulus? I would have tested the ability of the compounds to counteract the oxidative stimulus in milder conditions, for example with a H2O2 concentration causing a 50% cell toxicity instead of 90%. Can they explain this choice?

We agree with the reviewer. However, subtoxic levels of hydrogen peroxide induced inconsistent effect on HCE cell viability and could not be used.

4)     Statistical analyses are missing in all graphs. Authors should add an adequate statistical analysis to properly highlight differences among the various compounds.

Thank you, we agree, and statistics were included as follows:

In figures 3 and 4 the number of observations is only two, as was written in the figure captions. Therefore, the statistical analyses do not make sense in those cases.

In figure 5 there is a significant difference in ORAC values between 3-carene and S-limonene compared to pinenes and limonene. The results of the one-way variance analysis (ANOVA) with Tukey´s comparisons are now indicated with letters above the bars in the figure 5 and the results are listed on the supplementary Table S1.

Similarly, one-way variance analysis was done for the figures 6 and 7. The incubation time 50min was chosen to describe the differences between the treatments. The results of the statistics from the biosensor measurements are listed in Supplementary Tables S2-S9.

5)     For what concerns section 4, Materials and Methods, please create separate subsections for the various experiments and avoid long descriptions of the principles of the methods used, just the details of the experiments performed to replicate them.

The Materials and methods section was revised according to suggestions. Please see the pages 11 and 12.

6)     In the Conclusions section (page 12, lines 388-390), the authors state that “we show that the known evaporated amount of monoterpenes ….. released from wooden construction materials is thousand times lower than the harmful level for ocular epithelial cells”, however there is no reference to the literature that allow them to perform such comparison. I would suggest to move these sentence from the conclusion and add it to the discussion to develop it better and explain how the authors are able to say that the monoterpenes tested are not toxic at the concentrations released from softwood based on the tests performed in their manuscript.

There are three references included in the manuscript, where the monoterpene emission levels have been studied and why we think that the harmful contents of monoterpenes evaluated in the present study are not easily reached in indoor air. In discussion it was written that:

“In HCE cultures both α-pinene and 3-carene were cytotoxic and induced IL-6 production at the concentration of 0.5 mg/ml. In a previous study, the highest 3-carene emission was 0.00003 mg/mL from the fresh Scots pine blocks [4]. Very high content of terpenes can be harmful for eyes, but according to the present experiment the cytotoxicity level is thousands of times higher than that content evaporated from the indoor wood material [23] [3] [4].”

This observation is repeated in the conclusion section. However, we changed the conclusion slightly based on the suggestions of the evaluators.

[3] Hyttinen, M.; Masalin-Weijo, M.; Kalliokoski, P.; Pasanen, P. Comparison of VOC Emissions between Air-Dried and Heat-Treated Norway Spruce ( Picea Abies), Scots Pine ( Pinus Sylvesteris) and European Aspen ( Populus Tremula) Wood. Atmospheric environment (1994) 2010, 44, 5028-5033.

[4] Muilu-Mäkelä, R.; Kilpeläinen, P.; Kitunen, V.; Harju, A.; Venäläinen, M.; Sarjala, T. Indoor Storage Time Affects the Quality and Quantity of Volatile Monoterpenes Emitted from Softwood Timber. Holzforschung 2021, 75, 945-956.

[23] Wolkoff, P.; Nielsen, G.D. Effects by Inhalation of Abundant Fragrances in Indoor Air - an Overview. Environment International 2017, 101, 96-107.
